# STATIC, INITIALIZATION-BASED LAYER-WISE LEARNING RATES

## ABSTRACT

A major characteristic of the Adam optimizer is its adaptive step size modification, which prevents large gradients from dominating the update step size. Given that the simplicity and computational efficiency of first-order methods are a significant advantage for large-scale training, we investigate an extreme form of step size modification that assigns static, layer-wise learning rates inversely to the initial gradient magnitude. We observe this simple heuristic is surprisingly effective in improving the rate of convergence on LLM style models over eight contemporary optimizers, suggesting the possibility of a static, initialization-based preconditioner.

## 1 INTRODUCTION

There is considerable interest in improving training of Transformers (Vaswani et al., 2017) due to their ability to handle large datasets with scale. Originally designed for natural language processing, their capability to train large datasets at scale has made them a popular choice for other tasks such as vision (Dosovitskiy et al., 2021), which were previously dominated by convolutional neural networks. The success of Large Language Models (LLMs) trained on a large corpus of data further incentives the desire to improve transformer training, and training GPT-2 style Large Language Models (Radford et al., 2019) on autoregressive language modeling has emerged as a key benchmark (Semenov et al., 2025; Wen et al., 2025) to evaluate and demonstrate the superiority of various promising optimizers over Adam (Kingma & Ba, 2015).

Muon (Jordan et al., 2024) performs efficient orthogonalization to the update matrix, which is generated by SGD with momentum, and demonstrates impressive performance over the Adam baseline. AdEMAMix (Pagliardini et al., 2025) introduces an additional exponential moving average (EMA) of the gradients with high a $\beta_3$ to Adam to avoid the pitfall of only relying on recent gradients. SF-Adam (Defazio et al., 2024) is a schedule-free variant of Adam that maintains separate locations where the gradient is obtained, the update step is performed and is used for evaluation. Taniguchi et al. (2024) modifies the order of the momentum update and second-moment normalization to address the non-convergence problem of Adam (Reddi et al., 2018) without setting $\beta_2$ in a problem dependent manner. A common characteristic shared by these and many more optimizers is their dynamic nature: they rely on local, in-training time gradients as the guiding signal to determine the step size and direction.

On the other hand, the currently widely used Transformer archetype is a cumulative result of intricate engineering efforts. Some examples of the architectural engineering are the careful location of normalization layers (Xiong et al., 2020; Dehghani et al., 2023), type of normalization layers (Ba et al., 2016; Zhang & Sennrich, 2019), choice of activation function (Hendrycks & Gimpel, 2016; Ramachandran et al., 2017), and even the numerical values used to scale the residual branches (Noci et al., 2022; Yang et al., 2024; Bordelon et al., 2024) and the embedding layer (Takase et al., 2023). From an optimization perspective, Adam is known for its continued status as the default optimizer despite the best efforts, and informal conjectures have even suggested implicit tuning of architecture around Adam as a potential reason (Orabona, 2020). This leads to the question of whether architectural characteristics of the Transformer archetype can be used as further indicators to improve optimization of neural networks alongside general assumptions.

Dynamic methods that aim to better incorporate local gradient information is a well explored area that faces the challenge of minimizing the additional computation overhead. In contrast, static meth-

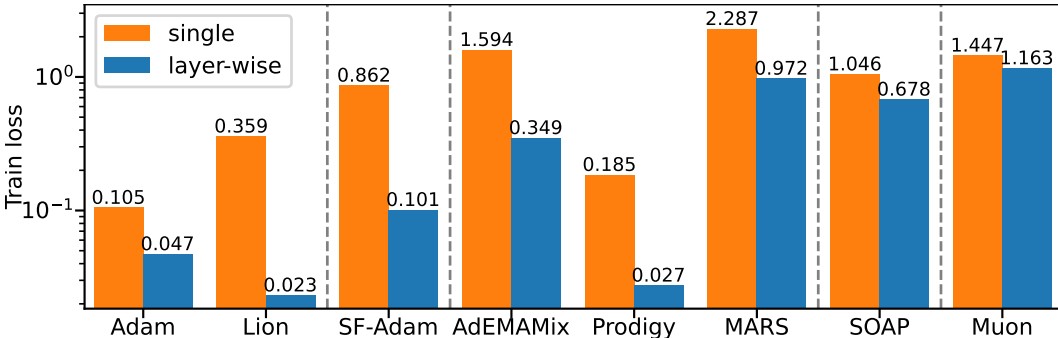

Figure 1: Train loss (↓) of 350M model on over-training regime with various contemporary optimizers. Trained for 1000, 1400, 600, 300, 150 iterations from left to right. Static, initialization-based learning rates improve train loss in all configurations.

ods remain largely unexplored despite their computation efficiency. We investigate assigning static, layer-wise learning rates inversely to the initial gradient magnitude as a method to account for implicit global information that is not captured by local gradients. The approach has similarities to the normalization performed in sign descent (Bernstein et al., 2018) and Adam in that layers with large initial gradients are assigned smaller update steps. The difference is that it is static and not dynamic, and that it is determined at initialization, a setting often used to investigate architectural influences such as layer depth, width and connectivity (Poole et al., 2016; Schoenholz et al., 2017; Balduzzi et al., 2017; Lee et al., 2019). As a static modifier that does not change during the entire training duration, it is of the least computationally intrusive methods to adjust optimization, making it an important candidate to compare and combine with other dynamic methods.

We evaluate on large model, small data setting (over-training) and small model, large data setting (standard) as indicators of training stability and performance of large model, large data settings. The layer-wise variants of eight contemporary optimizers demonstrate improved rate of convergence on 124M and 350M LLM models compared to their baseline versions on the over-training setting, while remaining competitive on the standard setting. The consistent results over a wide range of optimizers provide evidence for a global structure that is difficult to account for with current methods that rely heavily on local gradients. It also suggests the possibility of static, initialization-based preconditioning as an effective method for improving neural network training.

## 2 RELATED WORK

Adam (Kingma & Ba, 2015) style adaptive step size scaling has been successful in training neural networks, and Adam is often justified as a diagonal approximation of Adagrad (Duchi et al., 2011). Adagrad forms a preconditioner from the covariance matrix of accumulated gradients and transforms the update matrix using the preconditioner before taking a step to improve convergence properties. A consensus has yet to emerge regarding the significant aspects of Adam that contribute to its success, and various properties continue to be a topic of interest (Kunstner et al., 2023; Zhang et al., 2024; Xie et al., 2025a). The sign gradient descent nature of Adam, which makes the step size invariant to the gradient magnitude, is suggested to be a key factor for its superior performance over SGD in language tasks due to its ability to better handle heavy-tailed class imbalance by Kunstner et al. (2024). A variant of sign gradient descent with momentum was obtained through evolutionary search by (Chen et al., 2023), and sign-based methods offer advantages in scalability due to their lower memory overhead (Zhao et al., 2025).

Many works have incorporated dynamic layer-wise elements by using the layer-wise gradient (Singh et al., 2015) or by performing layer-wise gradient normalization and weight norm scaling (Zhou et al., 2019; You et al., 2020; Bernstein et al., 2020). Various methods explored adapting second order information to improve neural network training (Martens & Grosse, 2015; Li, 2017; Gupta et al., 2018; Yao et al., 2021) and preconditioning methods have been demonstrated to have layer-wise structure (Martens & Grosse, 2015; Yao et al., 2021; Zhang et al., 2024). Minimizing the

compute overhead continues to be a major objective in efficiently incorporating second-order information (Liu et al., 2024; Vyas et al., 2025).

Another direction that effectively modifies layer-wise learning rates is through the use of additional scale factors multiplied to the weights. This is used in various reparameterizations of neural networks such as Neural Tangent Kernal (NTK) and Maximal Update parameterization (muP) (Jacot et al., 2018; Yang et al., 2021; Large et al., 2024). MuP has been used to facilitate training of large models through hyperparameter transfer across model sizes, but relies on intricate theoretical machinery and potentially restricting assumptions. Milsom et al. (2025) instead measures the change in output logits due to layer-wise parameter changes and aligns the larger model to the smaller base model. Such schemes involve models across various scales and additional hyperparameter tuning, albeit on the smaller base model, and more fundamentally it is questionable if restraining the dynamics of a large model to follow the smaller one would not affect its capabilities as a larger model.

Assigning static learning rates using initial gradients is effectively a form of initialization, a process predominantly performed on the weights. Initial weight initialization schemes were proposed as solutions to the exploding/vanishing activation/gradient problem in vanilla feedforward networks and remain in wide use (Glorot & Bengio, 2010; He et al., 2015). Subsequent works explored downscaling the branch weights of residual blocks according to depth to prevent increasing activation variance with depth (Zhang et al., 2019b; De & Smith, 2020). This downscaling can be introduced through multiplicative factors rather than just scaling down the initial weight values (Hayou et al., 2021; Noci et al., 2022), which extends the effect of residual branch downscaling. More sophisticated schemes, such as orthogonal initialization and preserving dynamical isometry, have been used to improve training neural networks (Saxe et al., 2014; Xiao et al., 2018; Bachlechner et al., 2021), but these properties are generally not enforced beyond the initial point. The distance from the current weight to the initial weight is used in parameter-free methods to modify the global learning rate (Carmon & Hinder, 2022; Ivgi et al., 2023; Defazio & Mishchenko, 2023), and this work investigates using the initial gradient to set static, layer-wise learning rates.

## 3 PRELIMINARY

Gradient descent is an iterative optimization algorithm used to minimize a loss function $\mathcal{L}(\theta)$ by repeatedly updating the parameters $\theta$ in the direction of the negative gradient. While a variety of its variants are used to train neural networks with highly complex loss landscapes, the basic rule is to move the parameters opposite to the gradient $g^k = \nabla_\theta \mathcal{L}(\theta)$ at each step $k$ as follows:

$$\theta^{k+1} = \theta^k - \gamma g^k, \tag{1}$$

where $\gamma \in \mathbb{R}_+$ is the global learning rate that controls the step size of all parameters in each update. Choosing appropriate learning rates globally or to individual parameters have significant effects on the convergence speed and the properties of the point to which the algorithm converges. Typically, well-known optimizers process the gradient before it is multiplied by the global learning rate, as shown below:

$$\theta^{k+1} = \theta^k - \gamma O(g^k), \tag{2}$$

where $O(\cdot)$ represents the optimizer-specific preprocessing. For simplicity, consider Stochastic Gradient Descent (SGD) without momentum and weight decay. The update rule for the parameters is represented as follows:

$$\theta^{k+1} = \theta^k - \gamma g^k, \tag{3}$$

where the step size is directly proportional to the gradient's magnitude. This can lead to very different step sizes with fluctuating gradients. In contrast, Adam has been shown to maintain consistent update magnitudes even with unbalanced gradients (Liu et al., 2020). The sign descent property of Adam, which makes its step size invariant to the gradient's magnitude, has been investigated by various works (Balles & Hennig, 2018; Bernstein et al., 2018). Sign descent can be represented as below:

$$\theta^{k+1} = \theta^k - \gamma \text{sign}(g^k), \tag{4}$$

The parameters $\{\omega_1, \cdots, \omega_L\} = \theta$ in a deep neural network can be naturally grouped into different parameter blocks according to their layer-wise structure, where $L$ is the total number of layers. In

this work, layer-wise learning rates are represented as additional multipliers $\eta_l$ multiplied to the global learning rate $\gamma$ for each parameter block $\omega_l$ corresponding to the $l$-th layer. The modified update rule is:

$$\omega_l^{k+1} = \omega_l^k - \gamma\eta_l O(g_l^k). \tag{5}$$

Assigning static layer-wise learning rates $\eta_l$ to each $l$-th layer can have significant long-term effects for sign-based methods, as it permanently modifies the relative step sizes of each layer to be different while maintaining the invariance to local gradients. It is also among the least computationally intrusive methods to adjust training and is easily implemented in popular frameworks.

## 4 STATIC, INITIALIZATION-BASED LEARNING RATES

The difficulty of optimization in deep learning is often attributed to its non-linear and non-convex nature of neural network loss landscapes. However, the apparent success of first-order methods demonstrates that a point with desirable properties can be reached despite the non-convex nature. This is particularly confusing, as the non-convexity would make local gradient information to be significantly disconnected from global gradients. This discrepancy implies that there may be underlying assumptions or simplifying structures that make optimization more tractable.

There is growing evidence that approximate second-order methods can outperform their idealized counterparts (Benzing, 2022; Buffelli et al., 2024; Zhang et al., 2025; Xie et al., 2025b), which casts doubts on the general strategy of leveraging more sophisticated local gradient information. Similarly, the square root in Adam's preconditioner has complicated its classical interpretation as a preconditioner based on the empirical Fisher. Attempts to remove the square root in Adam have required additional mechanisms to preserve the update's scale-invariance with respect to the loss (Lin et al., 2024). These results suggest that the ingredient to improve optimization of neural networks may be difficult to obtain from more accurate or sophisticated processing of local gradients despite the additional computational overhead.

An alternative is a hypothetical global information that remains elusive from local gradients. AdE-MAMix (Pagliardini et al., 2025) introduces an additional EMA of gradients with a high $\beta_3$ to Adam to additionally incorporate older gradients to the update direction. It recommends a $\beta_3$ of 0.9999, which results in half of the gradient EMA to be spread over the previous 6,930 iterations. While this involves more global information compared to only relying on an EMA with $\beta_1$ of 0.9, which has a half life of 6.58, it is primarily concerned with the update direction and the step size normalization remains identical to Adam.

The opposite of dynamically tracking local information is to use a static value that does not change during training. The initial gradient is a convenient candidate for extrapolating global information, as the gradient at the end of training cannot be obtained beforehand, and local gradient information would already be leveraged by dynamic methods. The architecture, which is itself a result of evolutionary-like tuning around first-order methods, is also considered to be well-represented at initialization, making the initial gradient a natural candidate for extrapolating potential long-term dynamics.

Algorithm 1 outlines the proposed static layer-wise learning rate scheme. It begins from widely used standard weight initialization. Weights of linear layers are sampled from normal distribution $N(0, 1/f_{in})$, where $f_{in}$ is the fan-in number (Glorot & Bengio, 2010). As is standard practice, normalization scale layers are initialized to 1, and all bias layers, including the class token, position embedding, and relative position layers in Transformer architectures, are initialized to 0. The weight-tied embedding and head layer of language models is initialized based on the fan-in of the head layer, and makes is numerically similar to the heuristic of initializing to $\mathcal{N}(0, 0.02)$ (Shoeybi et al., 2019).

Next, the layer-wise gradient magnitude per parameter $G_l^T$ is used to inversely scale the relative layer-wise learning rates $\tilde{\eta}_l$ at line 11. It is collected while the weights are fixed at initialization, and we collect over $T$ batches for a more accurate measure. We use a $T$ of 100 as default, and measure over the batch as it ensures compatibility with batch norm. The decision to scale by the square root of the gradient and to not scale the normalization scale layers was determined empirically. Because learning rate is scaled by the raw gradient values, the relative values $\tilde{\eta}_l$ are normalized on a per-parameter basis to obtain the final layer-wise learning rates $\eta_l$.

---

**Algorithm 1** Static, initialization-based layer-wise learning rates

---

**Require:** Model, layer weights $\theta = \{\omega_1, ..., \omega_L\}$ and number of parameters $\{N_1, ..., N_L\}$
1: Set $G_l^0 = 0, \ l \in \{1, ..., L\}$
2: Initialize linear/conv layers to $\mathcal{N}(0, 1/f_{in})$, normalization scale layers to $\mathbf{1}$, bias layers to $\mathbf{0}$
3: **for** $t \leftarrow 1$ **to** $T$ **do**
4:    Sample minibatch from training set
5:    $\boldsymbol{g}_\theta^t \leftarrow \nabla_\theta \mathcal{L}^t(\theta)$
6:    **for** $l \leftarrow 1$ **to** $L$ **do**
7:       $G_l^t \leftarrow G_l^{t-1} + \frac{1}{N_l} \sum_{i \in \omega_l} |\boldsymbol{g}_i^t|$
8:    **end for**
9: **end for**
10: **for** $l \leftarrow 1$ **to** $L$ **do**
11:    $\tilde{\eta}_l \leftarrow \frac{1}{\sqrt{G_l^T}}$         {relative learning rates}
12: **end for**
13: $\tilde{\eta}_{avg} = \frac{1}{N_{sum}} \sum_{l=1}^{L} \tilde{\eta}_l N_l$      $\{ N_{sum} = \sum_{l=1}^{L} N_l\}$
14: **for** $l \leftarrow 1$ **to** $L$ **do**
15:    **if** $\omega_l$ is normalization scale layer **then**
16:       $\eta_l \leftarrow 1$
17:    **else**
18:       $\eta_l \leftarrow \frac{\tilde{\eta}_l}{\tilde{\eta}_{avg}}$
19:    **end if**
20: **end for**

---

There is a conceptual similarity between the static $\eta_l$ and the dynamic preconditioner of Adam/sign descent. Both methods prevent layers with large gradients from dominating the update step. In Adam the normalization is performed dynamically, whereas the static modifier $\eta_l$ "normalizes" layers that have large initial gradients to have smaller step sizes. The relative $\eta_l$ values are numerically similar to using the preconditioner of Adam based on the initial gradients and with an additional square root. A key distinction is that the existing square root in Adam is to the second moment estimate, whereas the square root for $\eta_l$ is performed over the gradient magnitude.

## 5 AUTOREGRESSIVE LANGUAGE MODELING

In this section we train LLMs on autoregressive language modeling on both large model, small data and small model, large data settings. Initial experiments with Adam and Lion (Chen et al., 2023) show the proposed static learning rates are most beneficial when used with $1/\sqrt{d_i}$ residual branch downscaling, where $d_i \in \{1, 2, ..., D\}$ and $D$ is the number attention modules. We extend the comparison to six additional contemporary optimizers and observe that the static layer-wise scheme consistently improves the rate of convergence across all optimizers in the large model, small data settings, which we refer to as the over-training regime.

**General experiment setup.** We experiment on 124M and 350M models adapted from nanoGPT style codebase Karpathy (2025). The models have an embedding dimension of 768 and 1024, respectively, and consist of 12 and 24 self-attention modules. It includes learned positional encoding, qk-normalization Dehghani et al. (2023) and RMSNorm Zhang & Sennrich (2019). Bias layers are not used and perform weight-tying of the embedding and head layer (Press & Wolf, 2017). All experiments begin training from the same fan-in used in Algorithm 1. We use the Fineweb (Penedo et al., 2024) dataset. We do not apply weight decay to avoid potential complication related to using different weight decay values per layer (Defazio, 2025). The absence of weight decay makes Adam identical to the widely used AdamW variant (Loshchilov & Hutter, 2019). Additional details and optimizer specific hyperparameters are reported in Appendix A.

### 5.1 OVER-TRAINING REGIME

**Experiment setup.** The over-training regime is where the model size is large compared to the data and is a convenient setting to verify improved convergence of various methods. This setting main-

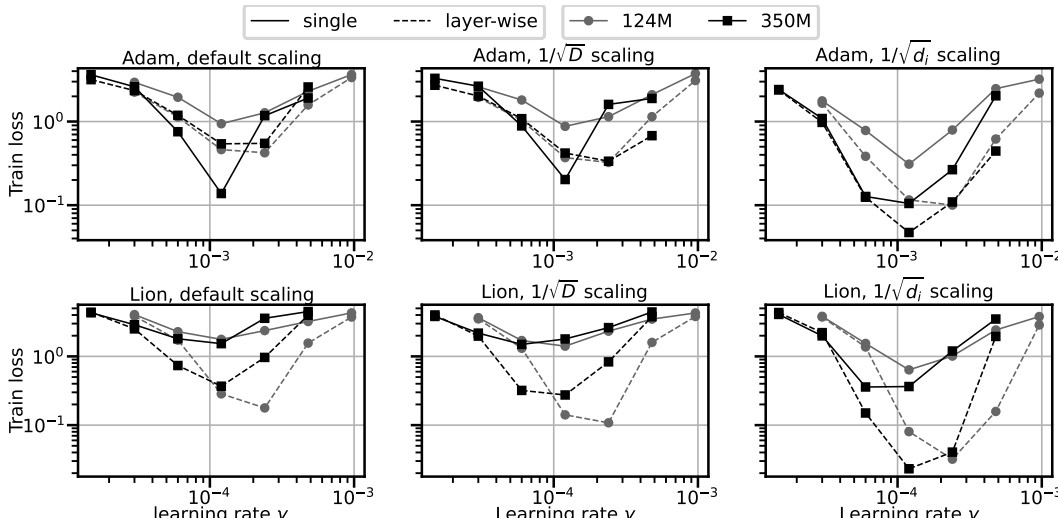

Figure 2: Train loss ($\downarrow$) on over-training regime with Adam (top) and Lion (bottom). Results from various depth scaling methods ranging from without depth scaling (left), with $1/\sqrt{D}$ scaling (middle) and with $1/\sqrt{d_i}$ scaling (right). The lowest train loss is achieved by the layer-wise learning rate with $1/\sqrt{d_i}$ depth scaling.

tains the non-linear and non-convex nature of deep learning optimization and can also be viewed as a proxy for predicting training instability that emerge at larger scales (Wortsman et al., 2024). We repeatedly iterate over 10 minibatches of data under a constant learning rate schedule. The number of iterations is 1000 for Adam and Lion, 1400 for SF-Adam, 600 for AdEMAMix, Prodigy (Mishchenko & Defazio, 2024), and MARS, and 300 for SOAP (Vyas et al., 2025) and 150 for Muon.

**Residual branch scaling.** We also investigate various forms of residual downscaling by the use of additional multipliers, as represented by $h(x) = x + \beta f(x)$. Such schemes have been investigated in various initialization works (De & Smith, 2020; Hayou et al., 2021; Noci et al., 2022) and for depth-wise hyperparameter transfer (Yang et al., 2024; Bordelon et al., 2024). We experiment with default downscaling of $\beta = 1$, downscaling of $\beta = 1/\sqrt{D}$ (Noci et al., 2022; Yang et al., 2024; Bordelon et al., 2024) and downscaling of $\beta = 1/\sqrt{d_i}$ that is similar to the Layernorm scaling suggested by Sun et al. (2025), where $D$ is the total number attention modules and $d_i \in \{1, 2, ..., D\}$.

**Adam and Lion.** Figure 2 shows the final train loss when sweeping over various global learning rates for Adam and Lion. Among the six settings {Adamw, Lion} $\times$ {1, $1/\sqrt{D}$, $1/\sqrt{d_i}$ }, the proposed layer-wise scheme improves train loss in all settings for Lion and with $1/\sqrt{d_i}$ depth scaling for Adamw. The lowest train loss for both optimizers is achieved by $1/\sqrt{d_i}$ depth scaling with layer-wise learning rates. Works that investigate the top singular value of residual Jacobians of fully trained ResNets (He et al., 2016) report it to scale inversely with depth (Rothauge et al., 2019; Li & Papyan, 2023), and our results reinforces the view that a depth-decreasing scaling (Zhang et al., 2019a; Hayou et al., 2021) is beneficial for improving trainability. Unless explicitly stated otherwise, remaining experiments are performed with $1/\sqrt{d_i}$ depth scaling.

**Observations.** The assigned learning rates are demonstrated in Figure 3, where the significantly larger values assigned to the query and key layers of the self-attention block are visible. This is similar to using an inverse temperature scaling of large values inside the attention softmax to counteract the small gradients of query and key layers (Noci et al., 2022), except that it is performed to all layers and not just the query and key layers. Additionally, the weight-tied embedding and head layer, represented as the first layer, is assigned relatively large values compared to most other layers. This aligns with the common practice of increasing the scale of embeddings (Dey et al., 2023; Hu et al., 2024). To verify all layers contribute to training we measure the EMA of (layer-wise) learning rate normalized weight updates in Figure 4. The average layer-wise updates are slightly higher for

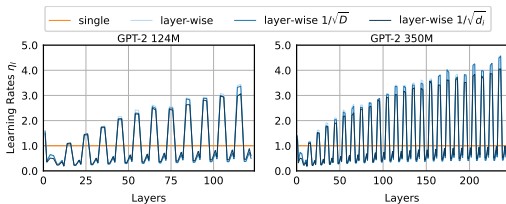

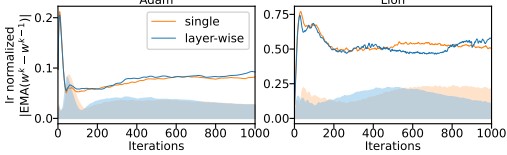

Figure 3: Assigned learning rates to 124M (Left) and 350M (Right) models.

Figure 4: Learning rate normalized EMA($\beta$=0.9) of layer-wise weight updates for 350M model on over-training regime for Adam (left) and Lion (right).

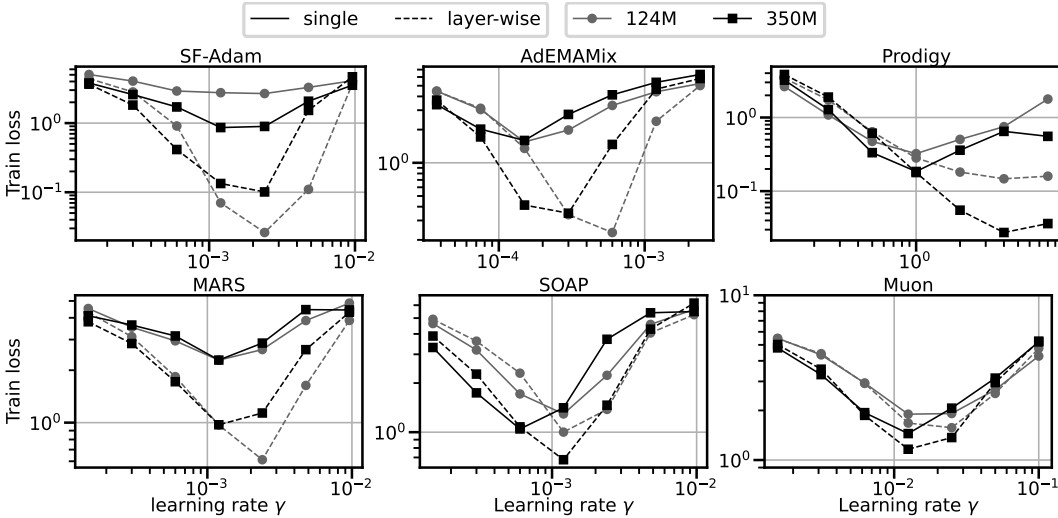

Figure 5: Train loss (↓) of LLM models on over-training regime. With $1/\sqrt{d_i}$ residual scaling. Layer-wise variants consistently achieve lower train loss.

Adam with similar variance, while Lion shows decreasing variance and an increasing update size in later iterations, verifying all layers contribute to training.

**Recent optimizers.** Figure 5 demonstrates the layer-wise scheme remains beneficial to recent optimizers, many of which are specifically designed for LLM pretraining. We note that different optimizers exhibited varying convergence speeds, thus requiring different training durations to clearly demonstrate the performance difference between the baseline and layer-wise variants. Figure 1 shows the lowest train loss achieved by each method on the 350M model. The consistent gains in both model sizes and all optimizer variations demonstrate the existence of a global structure that is not leveraged by methods relying solely on local gradients.

## 5.2 STANDARD SETTING

**Experiment setup.** This setting is the more common method used to evaluate various optimizers, where each batch of data is seen only once. Here the validation loss closely mirrors the train loss of the large dataset. We evaluate on the 124M model due to limited computational resources and the numerous possible combinations. We train with a cosine learning rate schedule that for a total of 18,865 iterations, which corresponds to 9.89 billion tokens.

**Adam and Lion.** Table 1 shows the validation loss across various residual scaling factors, where the layer-wise counterpart generally slightly outperforms the baseline variant. It can be seen the final performance in standard, large-scale setting does not necessarily correlate with the improvements in trainability observed in the over-training regime. This is somewhat expected, as the final performance may be a more complicated measure that is influenced by various factors that may conflict with training stability, similar to the traditional test-train generalization gap in over-training regimes.

Table 1: Validation loss ($\downarrow$) when training 124M model on standard setting with various depth scaling schemes.

| Optimizer | Single | | | Layer-wise | | |
|---|---|---|---|---|---|---|
| | 1 | $1/\sqrt{D}$ | $1/\sqrt{d_i}$ | 1 | $1/\sqrt{D}$ | $1/\sqrt{d_i}$ |
| Adamw | 3.3443 | 3.3374 | 3.3464 | 3.3495 | 3.3383 | 3.3376 |
| Lion | 3.3589 | 3.3423 | 3.3555 | 3.3439 | **3.3324** | 3.3368 |

We expect the benefits of improved trainability to become more apparent when scaling models to sizes where training stability is reported to be an issue (Zhang et al., 2022; Chowdhery et al., 2023).

Table 2: Validation loss ($\downarrow$) when training 124M model on standard setting with $1/\sqrt{d_i}$ residual scaling using recent optimizers.

| | SF-Adam | AdEMAMix | Prodigy | MARS | SOAP | Muon |
|---|---|---|---|---|---|---|
| Single | **3.3430** | 3.4584 | 3.3361 | **3.2751** | **3.2449** | **3.2551** |
| Layer-wise | 3.3629 | **3.4358** | **3.3191** | 3.2896 | 3.2738 | 3.2671 |

**Recent optimizers.** Table 2 demonstrates that the layer-wise variant improves validation loss for many recent, contemporary optimizers on the standard setting. We tuned the global learning rate for both the baseline and layer-wise methods. This setting is a highly optimized benchmark on which many of these methods have been designed and tuned for. The layer-wise variant remains competitive despite being tuned to perform better on the over-training regime compared to other methods. We consider the dual evaluation on both over-training and standard settings to ensure training stability is also important, particularly when used to extrapolate performance at larger scales.

Table 3: Train and validation loss ($\downarrow$) of 8-bit Adam and Galore on 124M model on over-training and standard settings. 32-bit optimizer states for the embedding layer is used for 8-bit Adam only, explaining the improved performance on standard settings compared to Adam.

| | Adam | Adam-lw | 8-bit Adam | 8-bit Adam-lw | Galore | Galore-lw |
|---|---|---|---|---|---|---|
| Over-train | 0.3094 | 0.1156 | 0.3482 | **0.1099** | 0.1545 | 0.1585 |
| Standard | 3.3464 | 3.3376 | 3.3145 | **3.3020** | 3.5495 | 3.5998 |

**Compatibility with memory-efficient Adam.** Quantization of the optimizer state variables allow training of larger models on the same hardware, and Table 3 demonstrates that the layer-wise scheme is compatible with 8-bt Adam (Dettmers et al., 2022). Galore performs low-rank gradient projection for memory efficient training and has successfully demonstrated trainability of large models up to 7B parameters (Zhao et al., 2024), but results in significant lower validation loss in standard settings. Galore achieves lower loss compared to the default Adam in the over-training regime, demonstrating that performance on the over-training regime can be indicative of training stability at larger scales. As the layer-wise variant of Lion demonstrate large performance improvement on the over-training regime, it suggests layer-wise sign descent methods as a promising method for achieving efficient large scale training alongside quantized versions of Adam.

**Takeaways.** Identifying scalable methods is challenging due to the phenomenon where initially promising approaches may not translate to larger settings (Xiao, 2024), and makes verification require significant computational resources (Wortsman et al., 2024; Everett et al., 2024). The layer-wise variants demonstrate clear improvements in the rate of convergence in simplified settings where gains in trainability are easily verifiable, and does so without introducing and tuning additional hyperparameters. Similar settings have been suggested as a proxy for predicting training stability at scale (Wortsman et al., 2024). The improvements are observed over a wide range of contemporary optimizers that have been specifically designed to improve final performance, while making minimal modifications to the original model and update direction. We consider these promising signs the

benefits will transfer to larger scales. A comprehensive explanation of the architectural collapse to the current archetype is likely to provide further insights into the success and limitations of modern neural network training, and how to improve it.

# 6 IMAGENET-1K

**Experiment settings.** We also conduct experiments on ImageNet-1k classification task using basic Inception-style preprocessing and strong data augmentation. The base models are used without additional residual scaling, and we adopt the model and data processing implementation from Wightman (2019). The results presented are from a previous version of the algorithm, in which the learning rates of the normalization scale layers were also adjusted. More detailed settings and hyperparameters are reported in Appendix B.

Table 4: Final top-1 validation accuracy (↑) for ImageNet-1k classification. Without residual scaling. Average and standard deviation of 3 runs.

| Model | #Params | Optimizer | Data Augmentation | Epochs | Learning Rate | |
|---|---|---|---|---|---|---|
| | | | | | Single | Layer-wise |
| ResNet-50 | 25.56M | SGD | basic | 90 | 76.91±.12 | 77.03±.09 |
| | | | basic | 200 | 77.24±.08 | **78.08**±.06 |
| | | Adamw | basic | 200 | 76.57±.08 | **77.01**±.06 |
| | | | strong | 300 | 78.25±.20 | **78.75**±.08 |
| ViT-S/16 | 22.05M | SGD | basic | 300 | 69.01±.32 | **71.84**±.51 |
| | | Adamw | basic | 300 | 75.46±.36 | 75.42±.17 |
| | | | strong | 300 | 77.39±.43 | **78.10**±.17 |
| Swin-T | 28.29M | Adamw | strong | 300 | 79.89±.15 | 79.95±.09 |
| ConvNeXt-T | 28.59M | Adamw | strong | 300 | 80.26±.15 | 80.43±.03 |

**Results.** The layer-wise variant demonstrates significant improvement for ResNet-50 (He et al., 2016) with SGD and Adam and for ViT-S/16 (Dosovitskiy et al., 2021) with SGD. The improvements in Swin-T (Liu et al., 2021) and ConvNeXt-T (Liu et al., 2022) are relatively minor. One explanation for these differences is that architectural improvements reduce the need for improved training schemes. Another explanation would be that the architecture has been implicitly tuned to have favorable results when trained with a single global learning rate. The assigned layer-wise learning rates shown in Figure 9 reveal a clear depth-increasing trend for ResNet-50. Large values are assigned to query and key layers similar to self-attention in LLMs. The stem patch embedding, class token layer and position embedding layers of ViT-S/16 are assigned lower values, which is unlike the larger values assigned to the weight-tied embedding and head layer in LLM models.

# 7 ABLATION STUDY

Table 5: Ablation study on scaling the normalization layers on over-training regime without qk-normalization. Reporting the lowest train loss (↓) when sweeping learning rates.

| Model | Optimizer | Single | Lw, norm scaled | Lw, norm not scaled |
|---|---|---|---|---|
| 124M | Adamw | 0.6827 | 0.6119 | **0.3993** |
| 124M | Lion | 0.8993 | 0.5361 | **0.4835** |

**Algorithm design.** The effect of not scaling the normalization scale layers is demonstrated in Table 5. For models without QK-normalization trained in the over-training regime, not scaling nor-

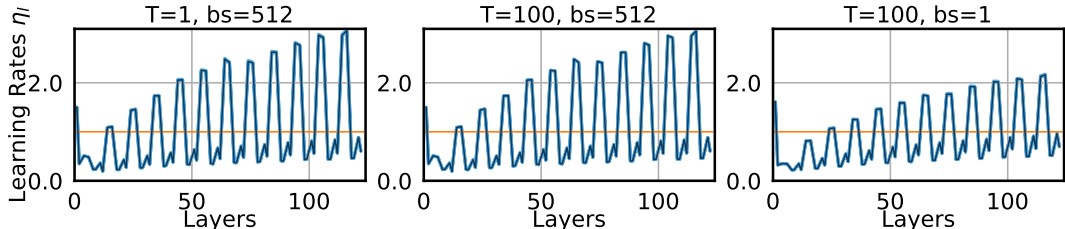

Figure 6: Mean and standard deviation of assigned learning rates to 124M model over 5 runs, with non-overlapping batches and same random initialization seed. The effect of T is barely noticeable but using a smaller batch size results in learning rates that are closer to 1.

malization layers significantly improves the training loss. The same trend is observed with QK-normalization, though it is less pronounced. Table 6 demonstrates that directly scaling by the layer-wise gradient, without performing the square root, results in a noticeable degradation in ImageNet-1k performance for ResNet-50. While preliminary testing showed it can very slightly improve train loss in over-training regime, we opt for a conservative approach to ensure generality.

Table 6: Ablation study on scaling with square root of gradient when training ResNet-50 on ImageNet-1k classification. Reporting top-1 validation accuracy ($\uparrow$). Normalization scale layers are not scaled.

| Optimizer | Epoch | Single | Lw, $1/\sqrt{G_l^T}$ | Lw, $1/G_l^T$ |
|---|---|---|---|---|
| SGD | 90 | 76.91$\pm$.12 | **77.06**$\pm$.14 | 76.11$\pm$.19 |
| Adamw | 200 | 76.57$\pm$.08 | **77.02**$\pm$.12 | 75.93$\pm$.02 |

**Hyperparameter sensitivity.** Figure 6 shows the effect of the hyperparameter T which is the number of batches from which to obtain the layer-wise learning rates. It can be seen that using a T of 1 shows essentially identical results to using a T=100, demonstrating it has negligible impact. However it can be seen that using a smaller batch size compared to the default 512 sequences results in similar but more mild learning rates. Table 7 demonstrates both layer-wise schemes result in similar train loss on over-training regime for Adam and Lion, which is expected they have similar trends.

Table 7: Train loss ($\downarrow$) of Adam and Lion in over-training regime when using learning rates from T=100, bs=512 and T=100, bs=1 on 124M model. Reporting mean and standard deviation of 5 runs.

| Adam | Adam-lw(bs=512) | Adam-lw(bs=1) | Lion | Lion-lw(bs=512) | Lion-lw(bs=1) |
|---|---|---|---|---|---|
| 0.3094 | 0.1122$\pm$.0089 | 0.1260$\pm$.0134 | 1.7695 | 0.0330$\pm$0.0031 | 0.0331$\pm$0.0028 |

## 8 CONCLUSION

We investigated a static, layer-wise learning rate scheme that prevents layers with large initial gradients from dominating the update step. Extensive experiments across eight optimizers in LLM pre-training and in vision tasks demonstrate the existence of a global structure in neural network training that is not leveraged by dynamic optimizers, suggesting the possibility of a static, initialization-based preconditioning method. It demonstrates that architectural factors play a critical role that requires further investigation for improved understanding of modern neural network training. A major limitation is the demonstration of correlation rather than causality between the initial layer-wise gradient and optimization, and promising applications are efficient large scale training and improving the performance of more diverse architectures through architecture-aware training.

## 9 REPRODUCIBILITY STATEMENT

Code is provided in the supplementary file.

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

## A  LLM EXPERIMENT HYPERPARAMETERS

For the optimizer hyperparameters, we primarily rely on the original recommended values or those suggested by Semenov et al. (2025), which were tuned in a slightly different setup. We avoid using large $\beta$ values because with a sequence length of 1024 and relatively large batch size of 512 results in fewer iterations compared to some other setups. The models have 12 and 16 attention heads and embedding dimensions of 768 and 1024. For standard settings we train with a warmup of 700 iterations. We use the GPT-2 tokenizer, which has a vocabulary size of 50,257. All training is performed using bfloat16 precision unless specified otherwise.

The results in Table 2 was obtained from extensive learning rate sweeps on the standard setting over power of 2 on both single and layer-wise learning rates, except for Prodigy as it is a learning-rate free method. Adam and Lion results in Table 1 was obtained using the optimal learning rates on the over-training regime demonstrated in Figure 2. Similar strategy was used for 8-bit Adam and Galore, and 8-bit Adam showed same optimal learning rate as Adam. For Galore we used a scale factor $\alpha$ of 1.0 as we found using 0.25 results in significantly shifted optimal learning rates in the over-training regime compared to Adam. The optimal learning rate of Galore in the over-training regime is 0.0024 for both single and layer-wise learning rate schemes.

Table 18 shows the performance of 124M model on the over-training regime. While weight decay slightly improves the single learning rate performance of SF-Adam and MARS, the general trend of the layer-wise variant achieving lower train loss remains identical. We use the common implementation of weight decay being multiplied by the global learning rate and the layer-wise learning rate if applicable. Weight decay is applied to all layers. Table 19 demonstrates the trend is identical with cosine learning rate schedule instead of constant in the overtraining regime. As the different schedule may affect the optimal learning rate we evaluated on $\times 0.5$, $\times 1.0$ and $\times 2.0$ the optimal value of the constant schedule.

Table 8: Adam hyperparameters for LLM training.

| Hyperparameter | Over-training regime | Standard setting |
|---|---|---|
| Learning rate | $0.0012 \times 2^{-3...3}$ | 0.0012 (s), 0.0024 (lw) |
| Number of warmup steps | 0 | 700 |
| Weight decay | 0.0 | 0.0 |
| Learning rate decay scheduler | none | cosine |
| Gradient clipping | 1.0 | 1.0 |
| Adam $\beta_1$ | 0.9 | 0.9 |
| Adam $\beta_2$ | 0.95 | 0.95 |

Table 9: Lion hyperparameters for LLM training.

| Hyperparameter | Over-training regime | Standard setting |
|---|---|---|
| Learning rate | $0.00012 \times 2^{-3...3}$ | 0.00012 (s), 0.00024 (lw) |
| Number of warmup steps | 0 | 700 |
| Weight decay | 0.0 | 0.0 |
| Learning rate decay scheduler | none | cosine |
| Gradient clipping | 1.0 | 1.0 |
| Lion $\beta_1$ | 0.95 | 0.95 |
| Lion $\beta_2$ | 0.98 | 0.98 |

## B  IMAGENET SETTINGS

Table 22 presents the hyperparameters used for ImageNet-1k training, many of which are based on values reported in Chen et al. (2023). We found that training with PyTorch's automatic mixed precision and bfloat16 improved ViT-S/16 performance compared to using float16 with loss scaling. We did not use gradient clipping or label smoothing. For strong data augmentation, we used a combination of RandAugment with $N = 2$ and $M = 10$ (Cubuk et al., 2020) and Mixup with strength of 0.5

Table 10: SF-Adam hyperparameters for LLM training. Bold hyperparameters give the best results.

| Hyperparameter | Over-training regime | Standard setting |
|---|---|---|
| Learning rate | $0.0012 \times 2^{-3...3}$ | 0.001, ... , **0.01** (lw), **0.02** (s), 0.04 |
| Number of warmup steps | 0 | 700 |
| Weight decay | 0.0 | 0.0 |
| Learning rate decay scheduler | none | cosine |
| Gradient clipping | none | 0.5 |
| SF-Adam $\beta_1$ | 0.9 | 0.98 |
| SF-Adam $\beta_2$ | 0.95 | 0.999 |

Table 11: AdEMAMix hyperparameters for LLM training. Bold hyperparameters give the best results.

| Hyperparameter | Over-training regime | Standard setting |
|---|---|---|
| Learning rate | $0.0012 \times 2^{-5...1}$ | 0.0005, **0.001** (s), **0.002** (lw) |
| Number of warmup steps | 0 | 700 |
| Weight decay | 0.0 | 0.0 |
| Learning rate decay scheduler | none | cosine |
| Gradient clipping | 0.5 | 0.5 |
| AdEMAMix $\beta_1$ | 0.8 | 0.9 |
| AdEMAMix $\beta_2$ | 0.95 | 0.999 |
| AdEMAMix $\beta_3$ | 0.995 | **0.999**, 0.9999 |
| AdEMAMix $\alpha$ | 8 | 8 |

(Zhang et al., 2018). All experiments were trained using a cosine decay learning rate schedule with warmup, image resolution of $224 \times 224$, and default momentum/beta hyperparameters for SGD and AdamW. Gradient accumulation was performed to simulate larger effective batch sizes. When adjusting the layer-wise learning rates the gradient was collected over one epoch which corresponds to a $T$ of 312~5004 depending on the batch size. Table 20 performs $\times 0.5$ and $\times 2.0$ learning rate sweep on the reported results for single learning rate for ResNet-50 and ViT-S/16, and demonstrate that the learning rate used is well tuned.

Table 21 demonstrates perliminary investigation on effect of layer-wise learning rates with AdaMuon, a Adam-style adaptation on the Muon optimizer (Si et al., 2025). The Adam-style adoptation allows the reuse of Adam learning rate hyperparameter as a starting point while simultaneously resolving the Adam-Muon step size discrepancy (Liu et al., 2025). The results demonstrate the layer-wise scheme can also benefit adaptive, Muon-based optimizers. We use a simplfied version that does not perform sign operation but maintains the nesterov momentum. We use $\beta_1$, $\beta_2$ of 0.9, 0.95 for Adam, and muon momentum of 0.95, which are the default recommended values for Muon. AdaMuon is applied only to the convolution layers including the first layer, while the remaining layers including the final linear layer are trained with Adam. We train for 35 epochs with a warmup of 8 epochs.

## C  USAGE OF LARGE LANGUAGE MODELS

Publically available LLMs were used for final writing checks and suggestions such as grammar checks, division of large sentences and usage of formal tone, some of which were incorporated by decision of the authors.

Table 12: Prodigy hyperparameters for LLM training.

| Hyperparameter | Over-training regime | Standard setting |
|---|---|---|
| Learning rate | $1.0 \times 2^{-3...3}$ | 1.0(s), 4.0(lw) |
| Number of warmup steps | 0 | 700 |
| Weight decay | 0.0 | 0.0 |
| Learning rate decay scheduler | none | cosine |
| Gradient clipping | 0.5 | 0.5 |
| Prodigy $\beta_1$ | 0.9 | 0.9 |
| Prodigy $\beta_2$ | 0.99 | 0.999 |
| use bias correction | yes | yes |

Table 13: MARS hyperparameters for LLM training. Bold hyperparameters give the best results.

| Hyperparameter | Over-training regime | Standard setting |
|---|---|---|
| Learning rate MARS | $0.0012 \times 2^{-3...3}$ | 0.006, 0.012, **0.024** (s), **0.048** (lw) |
| Learning rate Adam | $0.0012 \times 2^{-3...3}$ | 0.003, 0.006, **0.012** (s), **0.024** (lw) |
| Number of warmup steps | 0 | 700 |
| Weight decay | 0.0 | 0.0 |
| Learning rate decay scheduler | none | cosine |
| Gradient clipping | 0.5 | 0.5 |
| MARS $\beta_1$ | 0.95 | 0.99 |
| MARS $\beta_2$ | 0.99 | 0.99 |
| Adam $\beta_1$ | 0.9 | 0.9 |
| Adam $\beta_2$ | 0.95 | 0.95 |

Table 14: Muon hyperparameters for LLM training. Bold hyperparameters give the best results.

| Hyperparameter | Over-training regime | Standard setting |
|---|---|---|
| Learning rate Muon | $0.05 \times 2^{-5...1}$ | 0.05, 0.1, **0.2**, 0.04 |
| Learning rate Adam | $0.008 \times 2^{-5...1}$ | 0.008, 0.016, **0.032**, 0.064 |
| Number of warmup steps | 0 | 700 |
| Weight decay | 0.0 | 0.0 |
| Learning rate decay scheduler | none | cosine |
| Gradient clipping | 0.5 | 0.5 |
| Momentum Muon | 0.95 | 0.95 |
| Adam $\beta_1$ | 0.8 | 0.8 |
| Adam $\beta_2$ | 0.95 | 0.95 |
| Adam $\epsilon$ | 1.0e-10 | 1.0e-10 |

Table 15: SOAP hyperparameters for LLM training. Bold hyperparameters give the best results.

| Hyperparameter | Over-training regime | Standard setting |
|---|---|---|
| Learning rate SOAP | $0.0012 \times 2^{-5...1}$ | 0.002, ... , **0.032** (lw), **0.064** (s), 0.128 |
| Learning rate Adam | $0.0012 \times 2^{-5...1}$ | 0.002, ... , **0.032** (lw), **0.064** (s), 0.128 |
| Number of warmup steps | 0 | 700 |
| Weight decay | 0.0 | 0.0 |
| Learning rate decay scheduler | none | cosine |
| Gradient clipping | 0.5 | 0.5 |
| SOAP $\beta_1$ | 0.95 | 0.95 |
| SOAP $\beta_2$ | 0.95 | 0.95 |
| Preconditioning frequency | 10 | 10 |
| Adam $\beta_1$ | 0.95 | 0.95 |
| Adam $\beta_2$ | 0.95 | 0.95 |

Table 16: 8-bit Adam hyperparameters for LLM training.

| Hyperparameter | Over-training regime | Standard setting |
|---|---|---|
| Learning rate | $0.0012 \times 2^{-2...3}$ | 0.0012 (s), 0.0024 (lw) |
| Number of warmup steps | 0 | 700 |
| Weight decay | 0.0 | 0.0 |
| Learning rate decay scheduler | none | cosine |
| Gradient clipping | 1.0 | 1.0 |
| Adam $\beta_1$ | 0.9 | 0.9 |
| Adam $\beta_2$ | 0.95 | 0.95 |
| 32 bit state for embedding | yes | yes |

Table 17: Galore hyperparameters for LLM training.

| Hyperparameter | Over-training regime | Standard setting |
|---|---|---|
| Learning rate | $0.0012 \times 2^{0,1,2}$ | 0.0024 |
| Number of warmup steps | 0 | 700 |
| Weight decay | 0.0 | 0.0 |
| Learning rate decay scheduler | none | cosine |
| Gradient clipping | 1.0 | 1.0 |
| Adam $\beta_1$ | 0.9 | 0.9 |
| Adam $\beta_2$ | 0.95 | 0.95 |
| galore rank | 128 | 128 |
| galore subspace update frequency | 200 | 200 |
| scale factor $\alpha$ | 1.0 | 1.0 |

Table 18: Train loss ($\downarrow$) of 124M model on over-training regime with weight decay.

| wd-s/lw | AdamW | Lion | SF-Adam | AdEMAMix | Prodigy | MARS | SOAP | Muon |
|---|---|---|---|---|---|---|---|---|
| 0.0-s | 0.3094 | 0.6370 | 2.6748 | 1.5471 | 0.3228 | 2.6283 | 1.2935 | 1.8981 |
| 0.0-lw | 0.1003 | 0.0319 | 0.0261 | 0.2333 | 0.1472 | 0.6122 | 1.0009 | 1.5717 |
| 0.01-s | 0.3030 | 0.6562 | 1.2277 | 1.6105 | 0.3447 | 2.3080 | 1.5691 | 1.9043 |
| 0.01-lw | 0.1030 | 0.0304 | 0.0266 | 0.2654 | 0.1375 | 0.6004 | 1.0314 | 1.5546 |
| 0.1-s | 0.3683 | 0.6498 | 1.0449 | 1.5829 | 0.3780 | 2.3468 | 1.4939 | 1.9718 |
| 0.1-lw | 0.1164 | 0.0339 | 0.0250 | 0.2714 | 0.1300 | 0.6835 | 1.0437 | 1.5959 |

Table 19: Train loss ($\downarrow$) of 124M model on over-training regime with constant and cosine learning rate schedule.

| Lr sched | AdamW | Lion | AdEMAMix | Prodigy | MARS | SOAP | Muon |
|---|---|---|---|---|---|---|---|
| constant | 0.3094 | 0.6370 | 1.5471 | 0.3228 | 2.6283 | 1.2935 | 1.8981 |
| +lw | 0.1003 | 0.0319 | 0.2333 | 0.1472 | 0.6122 | 1.0009 | 1.5717 |
| cosine | 1.9571 | 3.2701 | 3.8427 | 0.2256 | 4.2393 | 1.5521 | 1.9037 |
| +lw | 0.2773 | 0.1912 | 1.2967 | 0.0122 | 2.5732 | 1.0560 | 1.5745 |

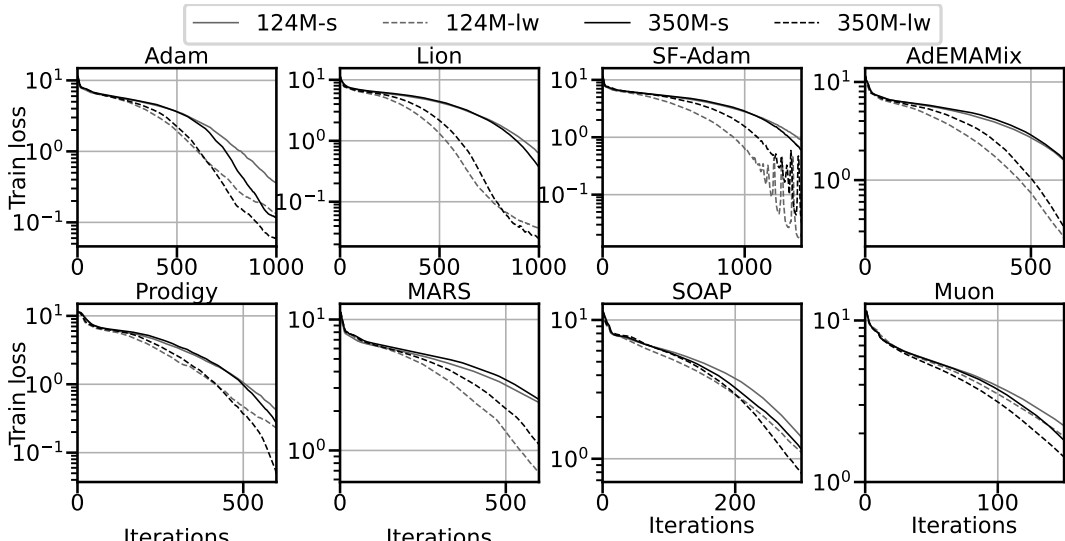

Figure 7: Train loss (↓) of LLM models on over-training regime across various optimizers. Demonstrating the best result across the learning rate sweep for each method. With $1/\sqrt{d_i}$ residual scaling. Single learning rate SF-Adam also exhibits spiking behavior when train loss becomes low by training for longer iterations.

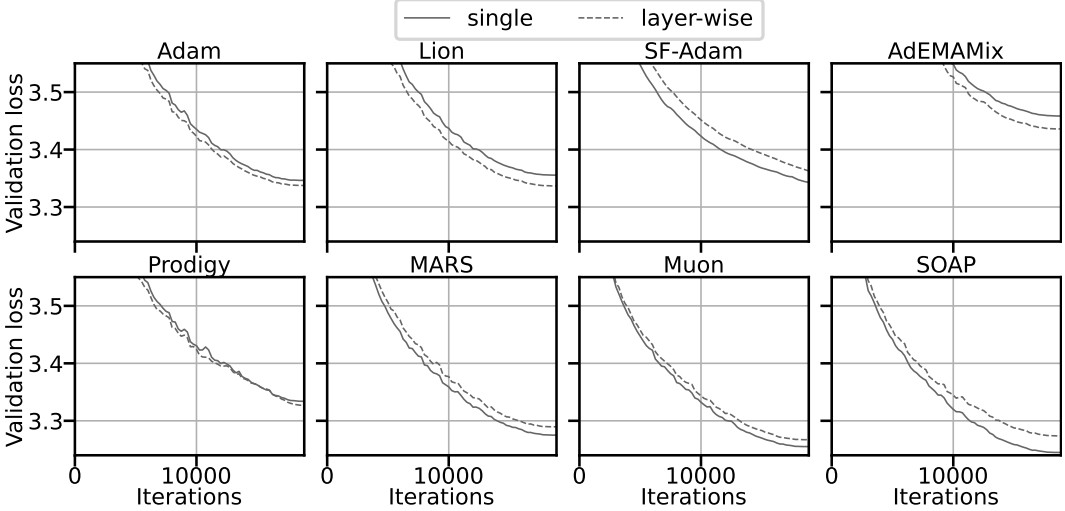

Figure 8: Validation loss (↓) of 124M model on standard setting across various optimizers. With $1/\sqrt{d_i}$ residual scaling.

Table 20: Final top-1 validation accuracy (↑) for ImageNet-1k classification for 0.5x, 2.0x the learning rate sweep with single learning rate. With basic augmentation. Reporting mean std of 3 runs when available, otherwise single run.

|  | Optimizer | Epochs | ×0.5 lr | ×1.0 lr | ×2.0 lr | ×1.0 lr-lw |
|---|---|---|---|---|---|---|
| ResNet-50 | SGD | 90 | 76.69 | 76.91±.12 | 76.30 | 77.03±.09 |
|  | AdamW | 200 | 76.11 | 76.57±.08 | 75.79 | **77.01**±.06 |
| ViT-S/16 | SGD | 300 | 67.30 | 69.01±.32 | diverge | **71.84**±.51 |
|  | AdamW | 300 | 75.37 | 75.46±.36 | 74.94 | 75.42±.17 |

Table 21: Final top-1 validation accuracy (↑) for ImageNet-1k classification on ResNet-50 with AdaMuon. With basic augmentation. Trained for 35 epochs. Without residual scaling. Normalization scale layers are not scaled.

|           | lr=0.003 | lr=0.006 | lr=0.012 |
|-----------|----------|----------|----------|
| Single    | 75.51    | 75.54    | 74.18    |
| Layer-wise| 75.65    | **75.92**| 75.48    |

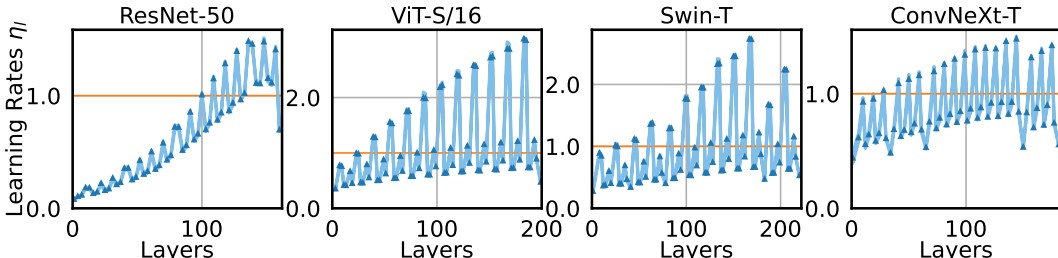

Figure 9: Assigned learning rates for ImageNet-1k classification. Only convolution/linear layers shown for visibility.

Table 22: ImageNet-1k hyperparameters

| Model | Dropout | Stoch Depth | Data Augmentation | Optimizer | Batch Size | lr | wd |
|-------|---------|-------------|-------------------|-----------|------------|-----|-----|
| ResNet-50 | - | - | basic | SGD | 256 | 0.1 | 1e-4 |
|  | - | - | basic | AdamW | 1024 | 3e-3 | 0.1 |
|  | - | - | strong | AdamW | 1024 | 3e-3 | 0.1 |
| ViT-S/16 | 0.1 | 0.1 | basic | SGD | 4096 | 0.8/1.6 | 1e-4 |
|  | 0.1 | 0.1 | basic | AdamW | 4096 | 1e-2 | 0.1 |
|  | - | - | strong | AdamW | 4096 | 1e-2 | 0.1 |
| Swin-T | - | 0.2 | strong | AdamW | 1024 | 1e-3 | 5e-2 |
| ConvNeXt-T | - | 0.1 | strong | AdamW | 4096 | 4e-4 | 5e-2 |

Table 23: ImageNet-1k real, v2 performance improvement due to layer-wise learning rate.

| Model | #Params | Optimizer | Data Augmentation | Epochs | Accuracy | |
|-------|---------|-----------|-------------------|--------|----------|----------|
|  |  |  |  |  | ReaL | V2 |
| ResNet-50 | 25.56M | SGD | basic | 90 | 83.63(+0.20) | 72.56(+0.33) |
|  |  |  | basic | 200 | 84.18(+0.77) | 73.24(+0.58) |
|  |  | AdamW | basic | 200 | 82.94(+0.43) | 72.24(+0.71) |
|  |  |  | strong | 300 | 85.24(+0.46) | 74.65(+0.47) |
| ViT-S/16 | 22.05M | SGD | basic | 300 | 77.96(+2.59) | 65.99(+2.85) |
|  |  | AdamW | basic | 300 | 81.44(-0.02) | 70.36(-0.24) |
|  |  |  | strong | 300 | 84.19(+0.82) | 73.61(+1.28) |
| Swin-T | 28.29M | AdamW | strong | 300 | 85.13(-0.06) | 74.98(+0.31) |
| ConvNeXt-T | 28.59M | AdamW | strong | 300 | 85.61(+0.33) | 75.98(+0.18) |

