# OpenReview forum: "Static, Initialization-based Layer-wise Learning Rates"
_ICLR.cc/2026/Conference — Submitted to ICLR 2026_

### Official Review · Reviewer_ync5 · 2025-10-17

**Soundness:** 2
**Presentation:** 1
**Contribution:** 2
**Rating:** 2
**Confidence:** 5

**Summary:**

The paper **"Static, Initialization-Based Layer-Wise Learning Rates"** proposes a novel optimization method that assigns **static, layer-wise learning rates** inversely proportional to each layer’s **initial gradient magnitude**. Unlike adaptive optimizers such as Adam, which dynamically adjust learning rates based on ongoing gradient statistics, this approach fixes layer-wise scaling factors at initialization. The authors empirically demonstrate that this static scheme consistently *improves convergence speed* across eight modern optimizers (including Adam, Lion, SF-Adam, AdEMAMix, Muon, SOAP, Prodigy, and MARS) when training large language models (LLMs) of 124M and 350M parameters and shows performance benefits on ImageNet-1k classification tasks with various architectures. The findings suggest that initialization-based preconditioning can capture global structural information missed by traditional gradient-based adaptive methods.

**Strengths:**

1. The proposed method requires no additional computation during training since the layer-wise scaling factors are computed once at initialization. This makes it lightweight and easily integrable into existing training pipelines while maintaining compatibility with diverse optimizers and architectures.

2. Experiments across multiple optimizers, model scales, and domains (language and vision) show consistent improvements in convergence and stability, supporting the claim that static, initialization-based preconditioning can serve as a general technique for enhancing neural network optimization.

**Weaknesses:**

1. The second paragraph of the introduction is confusing and should be moved to the related works section. It mainly provides a verbose discussion of previous studies that are only loosely connected to the main topic. Moreover, the last paragraph of the introduction largely repeats the content of this second paragraph, making the section redundant.

2. After reading the introduction, it remains unclear what the paper’s main methodology actually is. The authors should clearly define their proposed approach early in the introduction to help readers understand the core contribution.

3. The paper contains surprisingly sparse information. For instance, equations (1) through (5) appear almost identical, with only minor variations. The authors should consider grouping these equations and highlighting their differences more explicitly to improve clarity and facilitate comparison.

4. The first few paragraphs of Section 4 read more like related work than a direct motivation for the proposed method. For example, the discussion on “attempts to remove the square root in Adam” (line 185) appears tangential and does not clearly connect to the main argument of the paper.

5. The description of the proposed method is difficult to follow. To improve readability, key equations should be integrated directly into the main text instead of being referenced from the algorithm box without pointing to the line number. For example, in line 211, the statement “Next, the layer-wise gradient magnitude per parameter $G^T_l$ is used to inversely scale the relative...” should explicitly reference line 11 of Algorithm 1 to guide the reader.

6. The experiments presented are limited in scale and insufficient to support the paper’s claims. Given the surprising performance improvements reported, more extensive experiments are needed for validation. Prior work on optimizers [1] [2] [3] typically evaluates on larger-scale scenarios such as (1) pre-training on C4, (2) self-supervised fine-tuning on GLUE, (3) parameter-efficient fine-tuning on commonsense reasoning tasks, and (4) RLHF. Furthermore, scalability should be demonstrated on larger models (e.g., 7B parameters or more). In contrast, this paper only reports small-scale experiments on models up to 350M parameters, which limits the strength and generality of its conclusions.

Overall, while this work provides an interesting exploratory case study, it lacks sufficient methodological clarity and experimental rigor to be suitable for the ICLR main track.

## Reference
[1] GaLore: Memory-Efficient LLM Training by Gradient Low-Rank Projection, ICML 2024.

[2] APOLLO: SGD-like Memory, AdamW-level Performance, MLSys 2025.

[3] Adam-mini: Use Fewer Learning Rates to Gain More, ICLR 2025.

**Questions:**

1. What is the main motivation for studying static learning rates based on initialization gradient information? The paper should clearly explain why such an approach is valuable compared to existing adaptive methods.

2. How is the proposed layer-wise learning rate integrated with optimizers that already adapt to gradient magnitudes, such as Adam? Since Adam scales updates by the inverse square root of the gradient variance, the effects of the proposed method may overlap. Clarification on how these mechanisms interact is needed.

3. Is there any theoretical justification for why scaling by the initial gradient magnitude should improve optimization? Providing intuition or analysis on why this initialization-based scaling is effective would strengthen the paper’s contribution.

---

> ### Author Response · Authors · 2025-11-21
> **Response to Reviewer ync5**
>
> We sincerely thank the reviewer for providing valuable and insightful feedback. We address the main concerns below.
>
> **The experiments presented are limited in scale and insufficient to support the paper’s claims**
>
> How to address scalability is a difficult challenge that we believe cannot be supported through direct experiments alone. Even if a method directly demonstrate scalability on large models such as 7B parameters, a following question is whether it can scale to even larger models such as 32B parameters. The dual evaluation on large model, small data (over-training) and small model, large data (standard) settings is designed to isolate the effect of large model and large data into separate parts for more efficient investigation. This is in contrast to previous approaches which mostly focused on small model, large data setting and only used it as evidence for improved performance in larger settings.
>
> Additional results of Table 3 of Galore in the dual evaluation settings further demonstrate that Galore performs well in the over-training regime but does not perform well in the standard settings. It demonstrates the scalability of Galore scalability to 7B and limited performance due to low rank updates is consistent with the results obtained from the dual evaluation method.
>
> **main motivation for studying static learning rates based on initialization gradient information**
>
> The main motivation is the architectural collapse to the Transformer archetype that lacks comprehensive explanation, leading to the question whether architectural characteristics can be used as further indicators to improve optimization of neural networks alongside general assumptions. We have added the corresponding explanation in the introduction to make the motivation more clear.
>
> **Clarification on effects of the proposed method with optimizers that already adapt to gradient magnitudes**
>
> The static learning rates and adaptive methods are both similar and different, which makes it a promising direction to investigate whether the benefits of adaptive methods can be extracted in a similar but different form. A comprehensive explanation would require further insights into the role of architecture in facilitating the training of highly convex and complicated landscape of neural networks, which we believe is a promising direction for future work.
>
> **repetition of content in introduction and verbose discussion of previous studies that are only loosely connected to the main topic**
>
> We thank the reviewer for the suggestion, and have removed the discussion of previous studies in the last paragraph in the introduction. The discussion of previous studies serves to accurately position this work with relation to the many previous works, which we consider important given the subtle similarities and differences to methods that already adapt to gradient magnitudes.

---

> > ### Comment · Reviewer_ync5 · 2025-11-27
> >
> > We thank the authors for their responses and clarifications.
> >
> > Regarding the motivation, the framing should avoid overclaimed notions such as “architectural collapse.” The question of whether architectural characteristics can guide optimization is a broad topic involving many factors, including matrix-aware optimizers and the interaction between evolving architectures and the persistent AdamW training setup. The specific study of layer-wise learning rates is only one small component of this landscape and is only loosely connected to Transformer-specific behavior. Similar layer-wise learning-rate definitions could also be applied to MLP architectures.

---

> ### Author Response · Authors · 2025-11-27
>
> We thank the reviewer for the follow-up response and agree that this specific is study is only a small component of the landscape. We have modified the first and last sentence of the third paragraph of the introduction to remove the overclaimed notion and to make clear this study is more focused on the current Transformer archetype.
>
> We hope these resolve the concerns regarding overclaimed notions.

---

### Official Review · Reviewer_HDVK · 2025-10-31

**Soundness:** 3
**Presentation:** 3
**Contribution:** 2
**Rating:** 4
**Confidence:** 1

**Summary:**

This paper challenges the dominance of dynamic optimizers like Adam, which rely exclusively on local, in-training gradients to determine step sizes.
In particular, the authors argue that these dynamic-only methods fail to leverage a "global structure" evident in the model's initial state, which they claim can be captured by a layer’s gradient information at initialization across samples from the training set.
To leverage this information, the authors propose a scheme that assigns a static layer-wise learning rate proportional to $1 / \sqrt{G_l ^T}$, where $G_l ^T$ is the average gradient magnitude of the $l$th layer across $T$ training minibatch samples.
Experiments on LLM pretraining show this static initialization, when applied on top of eight different modern optimizers (including Adam and Lion), consistently improves the rate of convergence in an over-training regime.

**Strengths:**

1. This paper poses an interesting challenge to common assumptions in deep learning optimization

2. Experiments support the author's claims that static learning rates can be competitive with dynamic ones.

**Weaknesses:**

W1. While I appreciate the scientific investigation, the practical impact was not obvious to me. For everyday users of deep learning optimizers, what are the practical (+ theoretical) implications of an optimizer with static learning rates? In particular, just how significant is the dynamic overhead for Adam (and related methods) to justify a static one?

W2. The theoretical motivation of Algorithm 1 was not obvious to me, but this may be due to inexperience (see confidence).

**Questions:**

N/A

---

> ### Author Response · Authors · 2025-11-21
> **Response to Reviewer HDVK**
>
> We sincerely thank the reviewer for providing valuable and insightful feedback. We address the main concerns below.
>
> **practical (+ theoretical) implications of an optimizer with static learning rates**
>
> Currently the architecture has essentially collapsed to the Transformer architecture, and the static learning rates is a proof-of-concept that architectural factors can be used to improve training. The justification of a static over dynamic is more important conceptually rather than the practical overhead of dynamic methods, as we will explain below.
>
> The dual evaluation of large model, small data (over-training) and small model, large data (standard) setting demonstrates that while the standard setting is highly optimized, training stability indicated by over-training regime can be significantly improved by static learning rates. The additional results on Galore that demonstrate Galore performs well on over-training regime further reinforces this view. As a result, one immediate practical value is in improving trainability of large scale models, opening up the possibility of scaling up non-adaptive sign-based methods, as sign-descent methods demonstrates the most improvement in over-training regime. Admittedly this is may not be applicable to everyday users of deep learning optimizers.
>
> A potentially larger practical implication is the use of static learning rates to enlarge the availability of trainable architectures. As the Transformer architecture has been tuned around the use single learning rate, it is possible that the collapse to the transformer architecture is a result of the lack of incorporation of architectural bias into training. The proof-of-concept of incorporating architectural factors into training through static learning rates implies that with static learning rates, more diverse architectures may become more robust and trainable, leading to performance and efficiency gains across diverse areas in the long run. Convolution like architectures where the dimension shrinks or increases unlike Transformers is likely to benefit the most as it leads to significantly different gradient magnitudes across the layers, as demonstrated by experiments on ResNets. While limited by the demonstration of correlation rather than causality with respect to the initial gradients, we believe it is a promising direction with significant theoretical and practical implications. The theoretical implication would be that the collapse to the Transformer architecture requires further investigation to improve understanding of the success and limitations of modern neural network training.

---

> > ### Comment · Reviewer_HDVK · 2025-11-27
> >
> > Thank you for the clarification. This is quite interesting, then. I have raised my score to a 6. Good luck.

---

### Official Review · Reviewer_VRVo · 2025-10-31

**Soundness:** 3
**Presentation:** 3
**Contribution:** 3
**Rating:** 6
**Confidence:** 2

**Summary:**

This paper studies a simple but useful form of learning rate scheduling or preconditioning in the style of adaptive learning rate. The proposed method, STATIC, sets layer-wise learning rates at initialization, inversely proportional to each layer’s initial gradient magnitude, and keeps them fixed during training. This method can be combined with widely used optimizers with almost no extra computational cost. Experiments demonstrate the effectiveness of the method across multiple settings.

**Strengths:**

1) The paper studies a simple yet practical problem—layer-wise learning rate scaling that remains fixed during training—which is easy to implement and shows potential usefulness.
2) The motivation is clear, and the proposed method is reasonable. It is also optimizer-agnostic, making it suitable for improving a wide range of existing optimizers.
3) The experiments on both large language models (LLMs) and vision tasks are well presented and easy to follow.

**Weaknesses:**

1) The initial layer-wise learning rates are calculated by collecting gradients from $T$ minibatches. Therefore, the hyperparameter $T$ may impact performance. It would be better to justify this choice and include an ablation study. It is also important to examine whether the method is sensitive to the order of minibatches during this initialization phase.
2) In the standard training setting, the performance gain is limited—especially without depth scaling, as shown in Table 1. Even when combined with residual scaling $1/\sqrt{d}$, the gain remains modest compared to using residual scaling alone. This discrepancy should be discussed, particularly given the strong training performance observed in the over-training regime. Such discussion would directly impact the understanding and applicability of the approach.
3) Minor: As expected, the layer-wise scaling works particularly well with SGD (Table 3). This suggests that the method may be more beneficial for less adaptive optimizers compared to Adam. It would be useful to discuss potential application regimes where the method provides the most benefit. Additionally, if possible, consider discussing or testing the approach on memory-efficient approximations of Adam, such as Galore [1], 8-bit Adam and others, which are less adaptive and could provide further insight into the method’s scope.

[1] Jiawei Zhao, Zhenyu Zhang, Beidi Chen, Zhangyang Wang, Anima Anandkumar, and Yuandong
Tian. Galore: Memory-efficient llm training by gradient low-rank projection. arXiv
preprint arXiv:2403.03507, 2024.

**Questions:**

1) How do the results vary with different $T$ in the initial gradient calculation? and with shuffling or ordering of the initial minibatches?
2) Why is the performance gain significant in the over-training regime but limited in the standard training regime?
3) Minor: Could you also discuss or conduct experiments with memory-efficient approximations of Adam, such as Galore, to see whether the method benefits less adaptive variants of Adam?

Please see the [Weakness] section for more details.

---

> ### Author Response · Authors · 2025-11-21
> **Response to Reviewer VRVo**
>
> We sincerely thank the reviewer for the positive response and the valuable and insightful feedback. We address the concerns and questions below.
>
> **hyperparameter $T$ may impact performance**
>
> We have updated the draft to include hyperparameter sensitivity analysis. Figure 6 in the updated draft demonstrate that the assigned learning rates are rather incentive to the choice of $T$. We performed further analysis on the effect of the batch size when calculating the initial learning rates, and found that it results in more mild values when using batch size of 1. We note the choice of measure over the entire batch instead of individual data was in consideration of compatibility with batch norm, and included the additional explanation in the method. Table 7 of the updated draft demonstrate that the slight variation depending on the batch size does not result in measurable performance difference in the over-training regime.
>
> **limited performance gain in the standard training regime**
>
> While most works focus on that standard training regime for evaluation, we instead perform dual evaluation on both the standard and over-training regime. The standard training regime is a highly optimized benchmark in which many works have been tuned to perform well, which explains the limited gains on the standard training regime. Given that the scalability of the method is of high concern which many not be adequately represented in the standard training regime, we peform dual evaluation on the both large model, small data setting (over-training) and small data, large model setting (standard) to extrapolate performance on the large model, large data setting. As our approach aims to perform well on more settings, and results in significant performance gain in the over-training regime at a small trade-off in the standard regime when the learning rates are tuned to extreme values. We have added the corresponding description of the dual evaluation as indicators training stability and performance in the introduction and in the evaluation section.
>
> **the method may be more beneficial for less adaptive optimizers compared to Adam**
>
> We performed preliminary experiments on AdaMuon [1] optimizer on ResNet-50, ImageNet-1k classification by training for 35 epochs across various learning rate sweeps. The result demonstrate the method can also be beneficial to adaptive methods that incorporate both Adam-style adaptivity and Muon orthogonalization, and the results including the sweeps are in Table 21 in the Appendix.
>
> | Single | Layer-wise |
> | ----  | ---- |
> | 75.54 | **75.92** |
>
> **consider discussing or testing the approach on memory-efficient approximations of Adam, such as Galore, 8-bit Adam and others, which are less adaptive and could provide further insight into the method’s scope**
>
> Following the reviewer's suggestion, we have conducted experiments on 8-bit Adam [2] and Galore and included the results in Table 3 and as additional section of the updated draft. We present the results on 124M model below.
>
> While we find the layer-wise scheme does not seem to benefit Galore, 8-bit Adam shows promise in that it actually outperforms both Adam and Galore in terms both over-training and standard settings. For 8-bit Adam we use 32-bit optimizer states on the embedding layer, which explains the surprising performance increase compared to base Adam. Galore, which has successfully demonstrated scalability up to 7B model, improves upon Adam in the over-training regime but at a significant performance penalty at the standard setting, demonstrating that the over-training regime reflects trainability at larger scales. The results suggest layer-wise versions of 8-bit Adam and sign-descent methods (Lion) as promising direction for achieving efficient training of larger models while using less memory, as the layer-wise versions significantly improve performance in the over-training regime without sacrificing standard setting performance.
>
>
> | | Adam | Adam-lw | 8-bit Adam | 8-bit Adam-lw | Galore | Galore-lw |
> | ----  | ---- | ----  | ---- | ----  | ---- | ----  |
> | Over-train | 0.3094 | 0.1156 | 0.3482 | **0.1099** | 0.1545 | 0.1585 |
> | Standard | 3.3464 | 3.3376 | 3.3145 | **3.3020** | 3.5495 | 3.5998 |
>
> **Reference**
>
> [1] Si, C., Zhang, D., & Shen, W. (2025). Adamuon: Adaptive muon optimizer. arXiv preprint arXiv:2507.11005.
>
> [2] Dettmers, T., Lewis, M., Shleifer, S., & Zettlemoyer, L. (2021). 8-bit optimizers via block-wise quantization. arXiv preprint arXiv:2110.02861.

---

### Official Review · Reviewer_G3XZ · 2025-11-03

**Soundness:** 2
**Presentation:** 3
**Contribution:** 2
**Rating:** 4
**Confidence:** 4

**Summary:**

This paper assigns a per-layer learning rate multiplier computed from inverse of gradient at initialization, which is kept fixed throughout training. Results on image classification and language modelling show that this simple learning rate scheme can yield performance improvements on a variety of state-of-the-art optimizers including adam, muon, soap among others.

**Strengths:**

- The approach is simple: scale per-layer learning rates by inverse of gradient magnitude compute from average K batches at initialization and is kept fixed throughout training, so no extra memory or runtime overhead.
- Experiments on standard tasks (resnet-50 imagenet-1k, gpt-2 language modeling) to demonstrate the effectiveness of the approach and its compatibility with SOTA optimizers including adam, muon, soap, etc.

**Weaknesses:**

- Overtraining regime is constructed a bit arbitrarily using 10 batches of data and I'm not convinced that it's enough to reveal/accurately model the issues in overtraining regime. Probably just ablating over data in terms of chinchilla optimal ratio (1x = chinchilla optimal, 2-8x = over-training regime) could be a better proxy for reproducing issues in over-training regime (with a smaller model such as 30m in case of computational constraints).
- No clear improvements with layer-wise scheme in standard settings (language modelling 124m model with 10B tokens, table 1 and 2); similar trends in image classification. LR sweep values in appendix for standard setting look very limited (2-4 LRs, appendix section A) and some of these were directly borrowed from prior work which has different setup as mentioned in the same section. I believe a thorough sweep is critical for an empirical work like this.
- No weight decay is used which isn't the practice in realistic training setups. It should be part of the sweep and must be tuned for each optimizer separately.

**Questions:**

- Why is learning rate scheduler not used in over-trained setting (appendix A)?
- Can authors please clarify the number of sweep runs for baseline (single global LR) and the proposed approach (layer-wise LRs) in LM and imagenet experiments?

---

> ### Author Response · Authors · 2025-11-21
> **Response to Reviewer G3XZ (1/2)**
>
> We sincerely thank the reviewer for providing valuable and insightful feedback. We address the concerns and questions below.
>
> **Overtraining regime is constructed a bit arbitrarily using 10 batches of data and I'm not convinced that it's enough to reveal/accurately model the issues in overtraining regime**
>
> We would first like to clarify that we refer to the over-training regime as a setting where the number of data is small compared to the size of model. The regime the reviewer referred to is closer to what we call the standard setting, where the number of data is larger (4x) than the chinchilla optimal model size. To minimize confusion we will refer to the setting where number of data is much smaller as over-fitting regime in this response.
>
> The dual evaluation on over-fitting regime and standard settings is designed to decompose the large data, large model setting into small data, large model (over-fitting regime) and large data, small model (standard setting) such that investigation is more feasible. We chose 10 batches for over-fitting regime as it is within the range of the half life of exponential moving average of beta 0.9 and 0.95 (~6.57, ~14.5), and expect any number of batches that is much smaller than the chinchilla optimal (5000 for 124M) to yield similar results. The standard setting of language modeling 124m model with 10B tokens would correspond to the setting where the data size is much larger (4x) than the chinchilla optimal ratio.
>
> **No clear improvements with layer-wise scheme in standard settings (language modelling 124m model with 10B tokens)**
>
> While we are aware that evaluation on standard settings is considered the norm, it is also important to recognize that improvements on relatively small settings are known to not always transfer to larger settings. The CIFAR-ImageNet gap is one example and similar discrepancy would be expected when extrapolating from 124M model, 10B tokens.
>
> It is unrealistic to investigate large scale training settings which has large data and large models, and performing full sweep makes it even more infeasible. This necessitates decomposition of individual characteristics that together comprises large scale training to making investigation possible. As aforementioned, we decompose the large model, large data into small data, large model (over-fitting regime) and large data, small model (standard) setting.
>
> The reviewer is correct in that the layer-wise scheme is mainly beneficial in the small data, large model (over-fitting) setting which comes at a small price in the large data, small model (standard) settings. We consider it a reasonable tradeoff in that it improves "trainability" at a small cost of "capacity", particularly when using as basis to extrapolate performance at larger scales.
>
> **LR sweep values in appendix for standard setting look very limited**
>
> We thank the reviewer for the detailed feedback, and we found certain optimizers (sf-adam, soap) were indeed undertuned in standard settings. We have updated the paper with more thorough sweeps. The single learning-rate baseline of Table 2 the updated paper is result of extensive sweep including runs above and below the learning rate used for evaluation which  ensure the value is optimal. We have also identified and fixed a bug for Muon that applied Muon instead of Adam for the embedding layer when training with a single learning rate, and have updated the paper with the fixed result. This improves the single learning rate baseline but does not change the overall conclusion. We would like to note that for Prodigy we did not perform learning rate sweep on standard settings as it aims to be a learning-rate-free method that removes the need for learning rate tuning. The particular number of sweep depend on how early the optimal value was found, such that if the initial point is close the number of sweeps is lower.
>
> We would also like to clarify that Table 1 results are obtained by using learning rates obtained from the sweep in the over-fitting regime in Figure 2, and have updated the appendix with the corresponding description. Given the infeasibility of performing learning rate sweeps in large scale settings, it is a more realistic method for determining the hyperparameters. Nevertheless we performed extensive sweeps for Table 2 given the high interest in the optimal validation loss technically achievable for the 124M model, even if it the same procedure is infeasible in large scale training not only practically but also due to potential training stability issues that arise in scale.

---

> ### Author Response · Authors · 2025-11-21
> **Response to Reviewer G3XZ (2/2)**
>
> **LR sweep for image classification**
>
> We provide results for 0.5x and 2.0x learning rate for ImageNet-1k for the baseline single learning rates with basic data augmentation. While we report results of single runs, the performance is below or similar to the 1.0x learning rate demonstrating that the reported results in Table 3 reflect highly tuned versions for {ResNet, ViT-S/16}x{SGD, AdamW}. We have included the results in the Appendix and plan to include results for Swin-T and ConvNext-T in future revisions.
>
> | Model   | Optimizer | Epochs | Single (0.5x) | Single(1.0x) | Single(2.0x) | Layer-wise(1.0x) |
> | ----  | ---- | ---- | ----  | ---- | ---- | ---- |
> | ResNet-50 | SGD | 90 | 76.69 | 76.91±.12  | 76.30 |  77.03±.09   |
> | ResNet-50 | AdamW | 200 | 76.11 | 76.57±.08  | 75.79 |  **77.01**±.06   |
> | ViT-S/16 | SGD | 300 | 67.30 | 69.01±.32  | diverge  |  **71.84**±.51   |
> | ViT-S/16 | AdamW | 300 | 75.37 | 75.46±.36  | 74.94 | 75.42±.17 |
>
> **No weight decay is used which isn't the practice in realistic training setups**
>
> Weight decay was deliberately set to 0 as it may introduce additional complexity particularly if the values are not set optimally. That is, there exist a variety of methods to apply weight decay such as excluding it on normalization layers, excluding it on embedding layers only [1] and many more methods [2] [3]. Setting it to zero was decided to be a safe choice as most weight decay assigning schemes would become identical at 0.
>
> We performed additional experiments in the overfitting regime and report the results below. We use the common implementation where the weight decay as scaled by the global (and layer-wise) learning rate, and is applied to all parameters. Results demonstrate that the consistent trend remains identical across various weight decay values, and have included it in Table 19 in the updated draft.
>
> | wd-s/lw | AdamW | Lion | SF-Adam | AdEMAMix | Prodigy | MARS | SOAP |Muon |
> | ----  | ---- | ---- | ----  | ---- | ---- | ---- | ---- | ---- |
> | 0.0-s | 0.3094 | 0.6370 | 2.6748 | 1.5471 | 0.3228 | 2.6283 | 1.2935 | 1.8981 |
> | 0.0-lw | 0.1003 | 0.0319 | 0.0261 | 0.2333 | 0.1472 | 0.6122 | 1.0009 | 1.5717 |
> | 0.01-s | 0.3030 | 0.6562 | 1.2277 | 1.6105 | 0.3447 | 2.3080 | 1.5691 | 1.9043 |
> | 0.01-lw | 0.1030 | 0.0304 | 0.0266 | 0.2654 | 0.1375 | 0.6004 | 1.0314 | 1.5546 |
> | 0.1-s | 0.3683 | 0.6498 | 1.0449 | 1.5829 | 0.3780 | 2.3468 | 1.4939 | 1.9718 |
> | 0.1-lw | 0.1164 | 0.0339 | 0.0250 | 0.2714 | 0.1300 | 0.6835 | 1.0437 | 1.5959 |
>
> **learning rate scheduler in over-trained setting**
>
> As the layer-wise scheme is an additional multiplier that is independent from the global learning rate, we consider it as rather independent from learning rate schedules. We performed experiments using cosine learning rate schedule using 0.5x, 1.0x and 2.0x the optimal learning rate from the constant scheme and present the results below. It can be seen that the general trend persists, and the gap often widens compared to a constant schedule.
>
> | Lr sched | AdamW | Lion | AdEMAMix | Prodigy | MARS | SOAP | Muon |
> | ----  | ---- | ---- | ----  | ---- | ---- | ---- | ---- |
> | constant | 0.3094 | 0.6370 | 1.5471 | 0.3228  | 2.6283 | 1.2935 | 1.8981 |
> | +lw      | 0.1003 | 0.0319 | 0.2333  | 0.1472  | 0.6122 | 1.0009 | 1.5717 |
> | cosine   | 1.9571 | 3.2701 | 3.8427 | 0.2256  | 4.2393 | 1.5521 | 1.9037 |
> | +lw      | 0.2773 | 0.1912 | 1.2967 | 0.0122 | 2.5732 | 1.0560 | 1.5745 |
>
> We have included all additional results in the Appendix
>
> **Reference**
>
> [1] OLMo, T., Walsh, P., Soldaini, L., Groeneveld, D., Lo, K., Arora, S., ... & Hajishirzi, H. (2024). 2 OLMo 2 Furious. arXiv preprint arXiv:2501.00656.
>
> [2] Xie, Z., Xu, Z., Zhang, J., Sato, I., & Sugiyama, M. (2023). On the overlooked pitfalls of weight decay and how to mitigate them: A gradient-norm perspective. Advances in Neural Information Processing Systems, 36, 1208-1228.
>
> [3] Kosson, A., Messmer, B., & Jaggi, M. (2024, July). Rotational Equilibrium: How Weight Decay Balances Learning Across Neural Networks. In International Conference on Machine Learning (pp. 25333-25369). PMLR.

---

### Author Response · Authors · 2025-11-21
**General Response Summary**

We thank all reviewers for their careful reading and helpful feedback. We have modified the paper according to the provided comments. Below is a summary of the changes to the main text.

- (Reviewer G3XZ) additional clarification regarding the dual evaluation on over-training and standard settings in the introduction
- (Reviewer G3XZ) LR sweep values have been extended for standard setting in Table 2. We have also fixed a bug that improves the performance of Muon, and have updated the results accordingly. It does not change the overall results.
- (Reviewer VRVo) Analysis on the sensitivity to $T$ and its impact on the over-training regime through Figure 6 and Table 6.
- (Reviewer VRVo) More detailed explanation on the observed performance discrepancy between over-training and standard settings.
- (Reviewer VRVo) Additional section on compatibility on memory efficient optimizers with Table 3.
- (Reviewer HDVK) Included the need for improved understanding of architectural factors for improved understanding of modern neural network training, and large scale training and exploring different architectures as promising applications on the conclusion.
- (Reviewer ync5) Highlighting the architectural collapse to the Transformer archetype that lacks comprehensive explanation, motivating the exploration of architectural indicators to improve optimization of neural networks in the introduction.

---

### Author Response · Authors · 2025-11-30
**Rebuttal Overview**

**Rebuttal summary**

We sincerely thank all reviewers and Area Chairs for their time and thoughtful evaluation.

We have addressed the main concerns of all reviewers in the rebuttal and the updated version as below.

- (Reviewer G3XZ) Clarification of the evaluation on over-training and standard settings (**W1 W2**) and ensuring thorough lr sweeps and other optimizer hyperparameter ablations (**W3 Q1 Q2**)
- (Reviewer VRVo) Additional analysis on the effect of the hyperparameter $T$ (**W1 Q1**), discussion on significance of the gains in the over-training regime and the limited gains in standard settings (**W2 Q2**) and additional results on 8-bit Adam and Galore that provide further insights on the methods scope and the evaluation of over-training and standard settings (**W3 Q3**)
- (Reviewer HDVK) Clarification of the motivation and the practical(+theoretical) impact in improving training stability of large scale training and enlarging the choice of reliable and robust architectures through architecture-aware training methods, which this study of static learning rates demonstrate (**W1 W2**).
- (Reviewer ync5) Exploring the potential use of architectural characteristics of the Transformer archetype as further indicators to improve optimization of neural networks alongside general assumptions as the main motivation (**Q1**), justified by the gap in practice and theory in the wide use of Transformer architetype that goes against the general assumptions from an optimization perspective (**Q3**). We also clarified that the evaluation on both over-training and standard settings instead of just the standard setting support the strength and generality of the conclusion, and that the additional experiments on Galore performed during the rebuttal further strengthens it (**W6**). We have also incorporated many of the various suggestions on presentation into the revised version (**W1~W5**).

In particular, we would like to highlight that the main concerns of reviewer G3XZ (thorough experiment hyperparameter sweep) and reviewers HDVK, ync5 (motivation and significance) has been thoroughly addressed in the rebuttal. The current version is reflective of the contents of the rebuttal.

**Reviewer Response Summary**

- Reviewer ync5 has requested to avoid the use of overclaimed notions which was introduced in the rebuttal as an effort to clearly convey the motivation of this work. We have revised the paper to the best of our ability to resolve such concerns.

- Given the clarification of the motivation and significance of the work, Reviewer HDVK has increased the score from 4 to 6.

The discussion has been frozen before we were able to receive response from Reviewer G3XZ and VRVo, and before Reviewer ync5's decision regarding the score.

We hope this summary has been helpful under the new process.

---

### Meta-Review · Area_Chair_TwaZ · 2025-12-22

**Summary:**

This paper introduces a static, layer-wise learning rates selection for LLM training. The static learning rates are computed using initial gradient information. Extensive experiments demonstrated the potential of the proposed method in improving the performance of popular algorithms in LLM training. However, this paper is ready to be accepted for the following reasons: (i) the motivation is not clearly presented. The learning rate schedule approaches used in current LLM training practice doesn’t cause much overhead. So it is not very clear why a static learning rate is needed. This point was not sufficiently discussed in the paper, as pointed out by several reviewers. (ii) the experiments don’t show improvements on standard training settings. The authors decompose the large-model, large-data setting to two smaller settings: large-model small-data (over-training), and small-model large-data (standard-training). The proposed method doesn’t show much improvement on the standard training setting, which caused questions if the claims in the paper are drawn appropriately. (iii) the method lacks theoretical justifications. The proposed method is quite heuristic. There is no theory about the proposed approach. The algorithm will benefit from even some explanatory justifications, but the paper lacks discussions on this.

**Reviewer Concerns:**

Addressed: Analysis on the sensitivity to T.

Outstanding: Motivation, and tests on large-model large-data setting.

**Reviewer Scores:**

The authors added more numerical tests to address some questions raised by the reviewers. They also added more clarification on the motivation. But this is not enough to fully motivate the proposed method. Moreover, the method doesn't show much improvement on standard training setting, which may limit its applicability in large training.

---

### Decision · Program_Chairs · 2026-01-26

Reject